# Transcriptome-Wide Gene Expression Plasticity in *Stipa grandis* in Response to Grazing Intensity Differences

**DOI:** 10.3390/ijms222111882

**Published:** 2021-11-02

**Authors:** Zhenhua Dang, Yuanyuan Jia, Yunyun Tian, Jiabin Li, Yanan Zhang, Lei Huang, Cunzhu Liang, Peter J. Lockhart, Cory Matthew, Frank Yonghong Li

**Affiliations:** 1Ministry of Education Key Laboratory of Ecology and Resource Use of the Mongolian Plateau & Inner Mongolia Key Laboratory of Grassland Ecology, School of Ecology and Environment, Inner Mongolia University, Hohhot 010021, China; jyy773839714@163.com (Y.J.); lijiabinooooo@163.com (J.L.); zyanan419@163.com (Y.Z.); huanglei_1996@163.com (L.H.); bilcz@imu.edu.cn (C.L.); lifyhong@126.com (F.Y.L.); 2Ministry of Education Key Laboratory of Herbage & Endemic Crop Biotechnology, School of Life Sciences, Inner Mongolia University, Hohhot 010021, China; yunyuntian412@hotmail.com; 3School of Fundamental Sciences, College of Sciences, Massey University, Palmerston North 4442, New Zealand; P.J.Lockhart@massey.ac.nz; 4School of Agriculture and Environment, Massey University, Palmerston North 4442, New Zealand; C.Matthew@massey.ac.nz

**Keywords:** comparative transcriptomic analysis, gene expression plasticity, differentially expressed gene, grazing adaptation, Calvin–Benson cycle, photorespiration, *Stipa grandis*

## Abstract

Organisms have evolved effective and distinct adaptive strategies to survive. *Stipa grandis* is a representative species for studying the grazing effect on typical steppe plants in the Inner Mongolia Plateau. Although phenotypic (morphological and physiological) variations in *S. grandis* in response to long-term grazing have been identified, the molecular mechanisms underlying adaptations and plastic responses remain largely unknown. Here, we performed a transcriptomic analysis to investigate changes in gene expression of *S. grandis* under four different grazing intensities. As a result, a total of 2357 differentially expressed genes (DEGs) were identified among the tested grazing intensities, suggesting long-term grazing resulted in gene expression plasticity that affected diverse biological processes and metabolic pathways in *S. grandis*. DEGs were identified in RNA-Seq and qRT-PCR analyses that indicated the modulation of the Calvin–Benson cycle and photorespiration metabolic pathways. The key gene expression profiles encoding various proteins (e.g., ribulose-1,5-bisphosphate carboxylase/oxygenase, fructose-1,6-bisphosphate aldolase, glycolate oxidase, etc.) involved in these pathways suggest that they may synergistically respond to grazing to increase the resilience and stress tolerance of *S. grandis*. Our findings provide scientific clues for improving grassland use and protection and identifying important questions to address in future transcriptome studies.

## 1. Introduction

Grassland covers about 40% of the total land area worldwide [1]. It plays a crucial role in ecological security by regulating the climate, conserving water resources, preventing wind and water erosion, and in the provision of forage for pastoral production [2,3,4]. Grazing is the most common land use in grassland regions. Unfortunately, because of long-term inappropriate use, human activities, and adverse natural factors (e.g., warming, drought, and pest damage), grasslands have been extensively damaged, resulting in serious ecological issues. In particular, 74% of the grassland in northern China has become degraded because of decades of over-grazing and the fact that the region is gradually becoming more arid. This has seriously threatened the survival and growth of the grassland vegetation, the local biodiversity, and the livelihoods of local herders. 

Grazing affects the availability of resources essential for plant growth, including nutrients, water, and light, while also influencing ground temperatures. The ability of plants to use these resources is altered accordingly. These changes result in the redistribution of materials and energy in plants, thereby affecting herbage growth [5]. Plant phenotypes are influenced by the impact of the environment on the formation of individual traits, including morphological, physiological, or behavioral responses [6,7]. Many previous studies have explored how plant phenotypes are affected by grazing disturbance [8,9]. Under continuous grazing disturbance, herbage usually exhibits dwarfism-related characteristics, including decreased height and short internodes, short and narrow leaves, stiff branches, small clumps and seeds, and shallow root distribution [10,11]. These traits ultimately decrease the biomass of an individual plant and the whole community. A comparison by Louault et al. [8] of the functional traits between plants in a grassland area that has been grazed for 12 years and plants in a grassland area with no grazing indicated that among 22 traits (e.g., leaf economy, root morphology, reproductive characteristics, and phenology), seven traits related to plant height were significantly affected by grazing. In sward-forming grasses, responses to grazing have been referred to as tiller size-density compensation, a phenomenon which operates to optimize sward leaf area index through a higher density of smaller shoots in conditions of more intense grazing and a lower density of larger shoots under laxer grazing. Under more extreme grazing intensity, both the size and density of shoots are reduced [12]. In clump- or tussock-forming grasses, a phenomenon similar to tiller size-density compensation can occur at the tussock level, with grazing altering the size and density of tussocks [13]. 

At the physiological level, grazing has been found to induce a series of changes in mesophyll cells, including changes to levels of expression and activity of photosynthetic enzymes, chlorophyll content, electron transport capacity, and hormone regulation [14,15]. More specifically, the physiological processes of the photosynthetic system, carbon assimilation capacity, and water use efficiency are substantially enhanced in grazed plants in response to defoliation disturbance and other stress conditions [16,17]. Moreover, several biological events, such as cell division, meristem elongation, leaf growth, and tillering, are promoted. These changes have been linked with recovery from damage incurred by grazing [15,18,19] while some physiological responses to grazing may not help plant growth. Increased photosynthetic activity can result in the over-accumulation of metabolic byproducts, such as reactive oxygen species and free radicals, and these can damage membrane lipid structures and disrupt water and ion homeostasis in plant cells [20]. However, during the regrowth of herbage after grazing, the accumulation of large amounts of osmoprotectants (e.g., proline and betaine) and the activation of the reactive oxygen scavenging system has been observed. Both could help to eliminate and/or minimize the toxic effects of the metabolic byproducts [15,21,22]. Additionally, the feeding behavior of livestock has been found to stimulate plants to produce large quantities of defensive compounds, such as terpenes, flavonoids, and alkaloids, which can decrease the palatability and nutritional quality of herbage to protect plants from further grazing and pest damage [23].

The plastic phenotypic response of an organism is mediated through the regulation of the transcriptome. In this process, gene expression is an intermediate molecular phenotype that links the genotype to environmental changes, while also linking diverse types of cells, tissues, and organs to express different phenotypes [24,25]. Gene expression changes in response to changing environmental conditions [i.e., gene expression plasticity (GEP)], are crucial for phenotypic plasticity and adaptive evolution [26,27]. For example, in *Saccharomyces cerevisiae*, altered levels of gene expression of many genes in the ergosterol biosynthesis pathway have been attributed to an adaptive lineage-specific response [28]. In a recent study of seven *Drosophila* species, 64% of the observed expression divergence was associated with adaptive changes driven by directional selection, and the adaptive gene expression was enriched in functional classes, including regulation, sensory perception, sexual behavior, and morphology [29]. Over shorter time frames, the modulation of gene expression has been found to be an effective regulator and indicator of the phenotypic status of organisms exposed to fluctuating environmental conditions. Examples of the latter include the response of *Mytilus* mussels to changing tidal conditions [30] and the plastic response of the fish *Fundulus heteroclitus* undergoing thermal acclimation [31].

*Stipa grandis* (Poaceae, 2n = 44) is a dominant species on the typical steppe of the Inner Mongolian Plateau, thus the responses of this species to over-grazing represent a major part of the ecosystem response to grazing disturbance in this region [32]. To date, several investigations have been conducted regarding this topic. These pioneering studies examined how grazing influences the succession and construction of the *S. grandis* community [33,34], as well as the *S. grandis* biomass, functional traits, physiological ecology characteristics, population genetic diversity, and gene expression profiles [35,36]. Although studies evaluating responses to grazing suggested different populations of *S. grandis* exhibit phenotypic divergence to varying degrees, the transcriptome changes underlying such changes remain largely unknown. One study compared the transcriptomes of *S. grandis* under conditions of overgrazing and non-grazing [37]. This work has laid the foundation for our more detailed investigation of differential gene expression and grazing response of *S. grandis*. To better understand the phenotypic changes in *S. grandis* populations with different grazing conditions, we sequenced, assembled, and compared the transcriptomes of *S. grandis* under four grazing intensities. We identified gene expression dynamics and differentially expressed genes (DEGs) under different grazing treatments, and identified transcriptional regulation of genes closely associated with the Calvin–Benson cycle (CBC) and photorespiration. We herein elucidate the inferred divergent gene expression response of *S. grandis* under different grazing conditions. The present study develops our understanding of the adaptability of *S. grandis* to grazing and will facilitate the identification of the molecular mechanisms associated with their morphological and physiological changes of *S. grandis* under long-term grazing effects.

## 2. Results

### 2.1. Sequencing Output and de novo Assembly

Approximately 10 gigabase of clean data at the Q20 level (an error probability of 1%) were obtained for each sample. The clean reads with the length of 100 bp for all samples were de novo assembled into 67,705–115,918 unigenes, with a mean length and N50 value of 906–1373 bp and 1243–2078 bp, respectively (Table 1). To obtain the non-redundant and unextendable assemblies, the assembled unigenes of the 12 samples were further clustered into 251,412 all-unigenes, with an average length of 1854 bp and an average N50 value of 2536 bp (Table 1). Nearly 90% of the paired-end reads for each sample were mapped back to their own de novo assembled transcriptome (Table 1). A BUSCO analysis revealed that of 303 conserved sequences in the eukaryotic database, 68.7–94.1% complete and 3–23% fragmented BUSCOs were identified in the assembled *S. grandis* transcriptomes (Appendix A).

### 2.2. Gene Quantification and Functional Annotation

An analysis of the coefficient of variation (CV) indicated the expression of 33,241 all-unigenes in each grazing treatment was highly reproducible (CV ≤ 0.5) (Appendix A). Thus, these all-unigenes were retained for further analyses (results of the correlation analysis of the retained genes are presented in Appendix A). A BUSCO analysis of the filtered dataset showed that of the 303 orthologs, nearly 86.2% complete and 3.6% fragmented BUSCOs were identified (Appendix A). The number of these all-unigenes with at least one sequence match based on a BLAST search of public databases was as follows: 31,029 (93.35%) in the Nr database; 30,460 (91.63%) in the NT database; 25,313 (76.15%) in the Swiss-Prot database; 22,071 (66.40%) in the GO database; 26,133 (78.62%) in the KOG database; and 26,583 (79.97%) in the KEGG database (Appendix A). Overall, the top five species with BLAST hits to annotated unigenes were *Brachypodium distachyon* (39.47%), *Aegilops tauschii* (16.65%), *Oryza sativa* (6.72%), *Hordeum vulgare* (6.69%), and *Oryza brachyantha* (3.47%). The GO annotation results indicated that in the three major GO categories, biological process, cell component, and molecular function, ‘cellular process’ (11,620; 35%), ‘cell’ (13,537; 40.7%), and ‘binding’ (11,381; 34%) were the dominant categories (Appendix A). Many of the identified transcripts were classified in the ‘biological process’ and ‘cell component’ categories, whereas only a few genes belonged to the ‘molecular function’ category (Appendix A). The KEGG pathway analysis indicated that the most enriched pathways were ‘global and overview maps’ (5223; 15.7%), ‘translation’ (2678; 8.1%), ‘signal transduction’ (1277; 3.8%), and ‘transport and catabolism’ (1397; 4.2%) (Appendix A).

### 2.3. Identification and Clustering of Differentially Expressed Genes

Among the 33,241 all-unigenes, a total of 20,173 transcripts showed significant differential expression (FDR ≤ 0.001 and the absolute value of log_2_Ratio ≥ 1) in pairwise comparisons of the four grazing intensities (highlighted in bold in Appendix A). There were 5885 DEGs in CK vs. LG; 6914 in CK vs. MG; 8841 in CK vs. HG; 3947 in LG vs. MG; 14,850 in LG vs. HG; and 9564 in MG vs. HG. Of these DEGs, 2357 had a mean FPKM value greater than 10.0 across all samples (Appendix A) and were retained for further analysis. More specifically, the number of the DEGs in each comparison was as follows: 736 DEGs in CK vs. LG; 979 in CK vs. MG; 800 in CK vs. HG; 307 in LG vs. MG; 1674 in LG vs. HG; and 1176 in MG vs. HG. The K-means clustering analysis assigned the above DEGs to 12 transcriptional clusters (Figure 1A and Appendix A), with more than 200 DEGs in clusters 1, 2, 7, 9, and 12. The KEGG pathway enrichment analysis revealed that many of the DEGs in clusters 1, 3, 4, 5, 6, 10, and 11 were associated with photosynthesis-related metabolic pathways. These pathways included ‘photosynthesis’, ‘carbon fixation in photosynthetic organisms’, ‘carbon metabolism’, ‘porphyrin and chlorophyll metabolism’, ‘photosynthesis-antenna proteins’, and ‘glyoxylate and dicarboxylate metabolism’ (Figure 1B). For the GO enrichment analysis, the relatively predominant GO terms included organelle (chloroplast, ribosome, mitochondrial, etc.) organization, response to stimuli (heat and light), and cellular processes (translation, protein folding, gene silencing, etc.) (Figure 1C).

### 2.4. Differentially Expressed Genes Related to the Calvin–Benson Cycle

Among the analyzed transcripts, 114 transcripts encoded 11 enzymes involved in the CBC, and 38 were DEGs identified by the paired comparisons of the four grazing intensities (Appendix A). The DEGs were divided into two categories based on their expression profiles (Figure 2). One group included the DEGs encoding ribulose-1,5-bisphosphate carboxylase/oxygenases (Rubiscos), 3-phosphoglycerate kinases (PGKases), glyceraldehyde-3-phosphate dehydrogenases (GAPDHases), transketolases (Tkases), and ribose-phosphate isomerases (RPIases), and the gene expression levels varied slightly from CK to LG but were up-regulated significantly in response to grazing pressure (MG to HG). The other group comprised DEGs encoding fructose-1,6-bisphosphate aldolases (ALDases), sedoheptulose-1,7-bisphosphatases (SBPases), phosphoribulokinases (PRKases), and ribulose-phosphate epimerases (RPEases). The expression levels of these transcripts were significantly down-regulated from CK to LG, relatively stable from LG to MG, and then sharply up-regulated from MG to HG. Additionally, the expression levels of the unigenes encoding ALDases, PRKases, and RPEases under CK and HG conditions were almost the same (except for Unigene48588 and CL16574.Contig28). Among these DEGs, those that were highly expressed, with mean FPKM values greater than 100.0 across all samples (Appendix A), encoded the following: four Rubiscos, two PGKases, four GAPDHases, five ALDases, one fructose-1,6-bisphosphatase (FBPase), one TKase, one SBPase, one RPEase, one RPIase, one PRKase, and one Rubisco activase (RCA) (Figure 2). The expression profiles revealed significant changes for several transcripts in the CBC in response to grazing (Figure 2). Among the PGKase-encoding unigenes, the CL3380.Contig6 and CL3380.Contig10 expression levels were up-regulated nearly 2.4-times from LG (141.49 and 75.71, respectively) to HG (339.03 and 184.49, respectively). Regarding the GAPDHases, Unigene64285 expression was up-regulated significantly from LG (141.73) to HG (356.98). For the RPIases, CL843.Contig1 and CL843. Contig10 were similarly expressed from CK to LG but exhibited the opposite expression trend from LG to HG. Among the ALDases, Unigene48588 and Unigene48595 expression levels changed by as much as 9.08-times (1099.92 vs. 121.14) and 5.29-times (1500.74 vs. 283.65), respectively. The expression of the RCA unigene (Unigene11360) was sharply down-regulated by approximately 27-times following the grazing treatments, reflecting the apparent negative regulation of this unigene (Figure 2).

### 2.5. Differentially Expressed Genes Related to Photorespiration

A total of 127 all-unigenes and 38 DEGs were identified, encoding 12 enzymes related to the photorespiratory pathway (Appendix A). The expression levels of the DEGs encoding one 2-phosphoglycolate phosphatase (PGLPase) (Unigene4982), five catalases (CATases) (Unigene20659, Unigene17925, Unigene11709, Unigene15777, and CL8442.Contig9), two serine hydroxymethyltransferases (SHMTases) (CL3053.Contig39 and CL3053.Contig41), three glycolate oxidases (GOXases) (CL95.Contig34, Unigene9650, and CL95.Contig47), and two glycine decarboxylases (GDCases) (CL1834.Contig3 and CL1834.Contig22) were down-regulated from CK to LG, after which they were relatively steady before being up-regulated from MG to HG (Figure 3). The expression levels of the glutamine synthetases (GSases) (CL3536.Contig26), aminomethyltransferases (AMTases), glutamate: glyoxylate aminotransferases (GGTases), and hydroxypyruvate reductases (HPRases) unigenes were down-regulated from CK to LG and then gradually up-regulated from LG to HG (Figure 3). The GOXase (Unigene9650), dihydrolipoamide dehydrogenases (DIDases) (CL12727.Contig15), and serine: glyoxylate aminotransferases (SGTases) (CL593.Contig32) expression levels were gradually up-regulated from CK to MG and then sharply down-regulated from MG to HG (Figure 3). Regarding the GOXase unigene (CL14974.Contig19) and five CATase unigenes (CL10067.Contig2, CL10067.Contig4, CL8442.Contig1, CL10067.Contig 5, and CL8442.Contig4), their expression levels were up-regulated from CK to LG, after which they changed slightly from LG to MG and decreased from MG to HG (Figure 3). The expression levels of the unigenes encoding SHMTase (Unigene49346) and SGTase (CL593.Contig33) were generally down-regulated from CK to HG (Figure 3). The DEGs for one PGLPase, two GOXases, one GGTase, two SGTases, and two SHMTases were relatively highly expressed, with mean FPKM values exceeding 100.0 across all samples (Appendix A). Additionally, Unigene4982 (PGLPase) expression was up-regulated nearly 3-times from LG (84.81) to HG (253.03) (Appendix A). Unigene9650 (GOXase) was highly expressed, with a MG expression level (443.96) that was about 3-times higher than the CK expression level (143.59) (Appendix A). The expression of the SHMTase-encoding unigenes (CL3053.Contig39 and CL3053.Contig41) changed by as much as 3.43-times (326.94 vs. 95.39) and 2.15-times (595.44 vs. 276.86), respectively (Appendix A). Among SGTase unigenes, the CL593.Contig1 and CL593.Contig32 expression level changes were similar from MG to HG, but the opposite expression trend was detected from CK to LG (Figure 3).

### 2.6. Experimental Validation

To validate the gene expression plasticity, nine genes that were differentially expressed under the four grazing intensities were selected for qRT-PCR analysis. These analyses confirmed similar trends in expression patterns identified by RNA-seq for all genes under the various grazing conditions, although there were differences in the expression of these genes between 2018 and 2019 (Figure 4).

## 3. Discussion

### 3.1. Gene Expression Plasticity Dataset for Stipa grandis under Different Grazing Intensities

Under grazing conditions, plants generally exhibit two phases of response: an initial stress response that occurs within a few days after herbivore foraging and a regrowth response that occurs for several weeks following grazing [5]. One study identified that genes related to stress responses in wounded *S. grandis*, including wound, drought, and plant immunity, responded significantly over 12 hours compared with non-grazed *S. grandis* [37]. Another found that genes involved in the cellular-antioxidant, apoptotic, and amino acid metabolism pathways of *Leymus chinensis* responded to grazing within a 24 hours period [38]. In the present study, to explore the gene expression patterns of the regrowth stage in *S. grandis* in response to various long-term grazing intensities, we collected plant samples two weeks after grazing from the experimental grazing fields and analyzed the effects of the grazing intensities on the *S. grandis* transcriptome. Due to gene expression being substantially affected by fluctuations under the environmental conditions, the following strategies were adopted to minimize the influence of an open environment on gene expression and to obtain an accurate transcriptomic dataset for our study.

First, we applied rigorous sampling practices. Grazing reportedly affects photosynthesis-related pathways, which are characterized by obvious dynamic diurnal changes [39]. Accordingly, we strictly controlled the sampling time, with all samples collected between 9:00 am and 11:00 am. Under light conditions, the genes encoding many key enzymes are abundantly expressed in plants. These enzymes have important roles related to the regulation of photosynthetic activities in diverse plant species [40,41]. For example, Rubisco constitutes 30–50% of the soluble proteins in C_3_ plant leaves [42]. Consistent with this expectation, Rubisco unigenes (CL2153.Contig7, CL2153.Contig1, and Unigene53513) were highly expressed in our field samples, with mean FPKM values of 3543.93 across the grazing intensities (Appendix A). 

We sequenced with high coverage three biological replicates for each grazing intensity. According to several quality assessment parameters of the assembled transcripts (i.e., number, average length, and N50), the mapping rate of clean reads of each sample to their assembled transcriptome (Table 1), and the BUSCO analysis (Appendix A), we can infer that the transcriptional sequences obtained were of high quality, integrity and validity, indicating our sample collection method was suitable for the complex environment in the field. 

Third, we applied multiple data filtering levels. Although an informative transcriptome dataset was constructed, several factors (e.g., variable splicing, incomplete assembly, tandem variations, and assembly bias) caused the merged transcriptome to contain an unusual number of transcripts. However, by filtering out the unigenes identified as not reproducibly expressed based on the coefficient of variation, a rational number (33,241) of unigenes was retained for subsequent analyses. This process not only significantly decreased the sample bias resulting from the assembly method, it also ensured the comprehensive coverage of the *S. grandis* transcriptome. Gene annotations are considered useful for assessing the accuracy of the transcript sequences assembled from short-read data [43]. Among the analyzed transcripts, 93.42% were annotated based on at least one of the screened public databases, and these transcripts were matched with a high probability score with homologs from model species, including *Brachypodium distachyon*, *Aegilops tauschii*, *Oryza sativa*, and *Hordeum vulgare*. This suggests that the *S. grandis* transcriptome in the present study was assembled and annotated correctly and that the unannotated transcripts probably represent genetic information unique to *S. grandis*. Our analyses indicated that 20,173 genes were differentially expressed across four grazing intensities. This number of DEGs was much higher than previously reported in a comparison of grazed and non-grazed pastures of *S. grandis*, wherein 13,221 DEGs were identified among 94,674 transcripts [37]. A lower number of DEGs were found in a comparison of over-grazed and non-grazed pastures of *L. chinensis*: 3341 DEGs identified in 116,356 unigenes [38], and an even lower number of DEGs were found in a comparison of ungrazed and grazed populations of *Stipa breviflora*: 686 DEGs obtained from 111,018 unigenes [44]. While the number of DEGs identified will be influenced by the analysis pipelines and filtering parameters used (please see comment below on DEGseq), the difference in the relative number of DEGs may also be explained by the greater environmental gradient across our four grazing conditions (Figure 5). Consistent with this interpretation, most DEGs identified in pairwise comparisons were inferred under the most contrasting environmental conditions: LG vs. HG (14,850 DEGs identified), followed by MG vs. HG (9564 DEGs identified), CK vs. HG (8841 DEGs identified), etc. To identify the most representative transcripts responding to grazing effects in *S. grandis*, an FPKM value greater than 10.0 across all samples was used as a threshold to further filter the overall identified DEGs. Accordingly, 2357 DEGs were identified, with functions suggesting they may have major roles related to basic metabolic activities and grazing responses in *S. grandis* (Figure 1B,C, Appendix A). Several GO terms (e.g., catalytic activity, metabolic process, and cell part) and KEGG pathways (e.g., ribosome, glyoxylate and dicarboxylate metabolism, carbon metabolism, photosynthesis, and carbon fixation in photosynthetic organisms) were enriched among these unigenes. This information provides some insight into the cryptic physiological, plastic and adaptive responses of *S. grandis* to recover under varying levels of grazing stress. Similar to the response of one-week grazed alfalfa (*Medicago sativa*) [45], patterns of differential expression of *S. grandis* suggested that carbon assimilation, ribosome organization, response to stimuli, and translation were at a more functional level following the recovery period after grazing. In contrast, cellular processes and metabolic pathways including secondary metabolite production, hormone signaling, and wound response, etc., were relatively inactive in *S. grandis*. This suggests that while the regulation of gene expression in response to grazing shows some differences between species, the regulation of photosynthesis may exhibit a common response to grazing pressures (Figure 1).

Lastly, potential shortcomings of the present study need to be mentioned. Gene quantification and differential expression analyses were performed using FPKM normalization and the R package DEGseq. Alternative methods that include normalization for both library size and transcriptome gene composition have been found to be much more effective than DEGseq in distinguishing true and false positives (e.g., DESeq2 and edgeR) [46,47,48]. However, when the number of biological replicates is small, such as in our study, DEGseq has also been found to produce far fewer false negatives than DESeq2 and edgeR [46]. This may have helped our study in which we had a small number of biological replicates for analysis. We found that gene and pathway enrichment on the initial set of differentially expressed candidate genes identified by DEGseq was effective in identifying true positives that were subsequently validated using qRT-PCR. The genetic differentiation of populations can also lead to false positives and misleading inferences of differential expression [49]. This was also not accounted for in our study. The impact of false positives and false negatives on downstream ontology inferences is an important issue for biologists that requires further study.

### 3.2. The Expression Patterns of Genes Related to the Calvin-Benson Cycle in Stipa grandis Were Changed under Different Grazing Intensities

Photosynthesis is a sensitive indicator of grazing stress in various grassland plants. The CBC is the initial pathway for photosynthetic carbon fixation, and changes to the expression of genes encoding the associated enzymes reportedly influence the growth of higher plants [40,50]. In the current study, grazing-induced transcriptional changes were detected for 114 unigenes encoding 12 enzymes involved in the CBC (Figure 2 and Appendix A), suggesting that various grazing conditions altered the expression patterns of these CBC-related unigenes.

The expression levels of CBC-related genes were compared in *S. grandis* between grazed plots (LG, MG, and HG) and non-grazing (CK) conditions, wherein the vegetation represents the top-level community [51,52]. Important to note is that herbivory is a form of predation in which animals draw off for their own use, energy and nutrients from the plants they graze, and that grasses in general are adapted to herbivory as a feature of their natural environment. Previous studies have shown that, with increased grazing intensity, the exposed soil surface area of plots increases, the canopy height of the plant community decreases, and the existing aboveground biomass (including the litter) decreases (Figure 5) [53]. It follows logically that a gradient of defoliation intensity such as in this experiment, also introduces a gradient of biomass removal intensity that will have impact on both the energy status and nutrient status of the plant. In the temperate sward forming forage grass *Lolium perenne*, carbohydrate levels fall rapidly after defoliation and recover gradually over approximately two weeks [54], while a phenomenon known as shoot size-density compensation comes into play such that a higher density of smaller shoots will help in the restoration of lost leaf area under more severe defoliation. In very severe defoliation, new shoots do not appear, likely because of substrate limitations [12]. More complex processes come into play in determining grazing effects on tussock-forming grasses [13], of which *S. grandis* is an example (Figure 5); however, it would be expected that more intense defoliation would impose greater substrate limitations in tussock-forming grasses like *S. grandis*, as in the sward-forming grasses. This principle of progressively reducing plant substrate status with increasing grazing intensity thus provides one framework against which to understand the photosynthesis responses observed in the present experiment. While the CBC performed similarly under non-grazing (CK) and heavy grazing (HG) conditions, the expression levels of CBC genes relevant for *S. grandis* growth and survival varied considerably between CK and HG. Specifically, the HG treatment induced the up-regulated expression of several unigenes encoding CBC enzymes. We propose that this response may be part of an adaptive strategy and plastic response that enables substrate-depleted or damaged plants to use their limited photosynthetic units to reconstruct organs and to maintain an appropriate balance in the materials and energy metabolism in the above- and below-ground plant parts [19,55].

This suggestion is consistent with a grazing optimization hypothesis that states that the unaffected biomass and small stature of plants under grazing stress reflect the promotion of net primary production [56]. Increased photosynthetic rate is a mechanism that has been suggested to support this hypothesis [57]. Furthermore, studies on the Inner Mongolia steppe have revealed that under frequent grazing stress, *S. grandis* plants exhibit dwarfism and induce efficient compensatory photosynthetic activities that can promote leaf regeneration and resistance to severe grazing [5,36]. This regenerative ability is crucial because *S. grandis* survival requires the rapid restoration of active photosynthesis and growth [58]. After grazing, the remaining or newly developed organs can undergo physiological changes that enhance photosynthesis, which can further increase the photosynthetic capacity of the grazed plants [55,59]. Under LG and MG conditions, when plant diversity is relatively high, livestock will selectively graze on the highly palatable vegetation, such as *L*. *chinensis*, *Cleistogenes squarrosa*, and *Chenopodium glaucum*, resulting in minimal damage to *S. grandis* [60]. Thus, the competition for resources decreases for *S. grandis* compared with the competition under CK conditions. The relatively abundant resources make it easier for *S. grandis* plants to maximize physiological growth under LG and MG conditions [5]. Therefore, it is unnecessary to invest as much material and energy in resource competition. In the present study, the expression levels of unigenes encoding Rubisco, SBPase, TKasse, and ALDase were down-regulated significantly under LG and MG conditions, which was consistent with the observed changes to photosynthetic-related physiology and phenotypic characteristics of *S. grandis* in response to grazing.

However, the above hypothesis of response to substrate depletion and herbivory damage does not explain the elevated levels of expression of CBC-related genes in the CK plot compared with those in the LG plot. In regions with a high plant density, vegetation with a large above-ground biomass, and rich biodiversity, competition is the main driving force of the community [61]. To gain a competitive advantage with limited resources in this situation, we propose *S. grandis* will increase its carbon fixation capacity to generate more resources. If this assumption is correct, we predict that key genes encoding CBC enzymes would be expected to be highly expressed to increase photosynthetic activity under conditions of increased competition.

Control over the rate of carbon fixation in the CBC is shared by a few enzymes. Analyses of antisense plants generated direct experimental evidence that expression-level changes to Rubisco, SBPase, ALDase, and TKase genes can influence the carbon flux through the CBC, with consequences for photosynthesis and growth [40]. In our study, of the identified unigenes in this cycle, 38 were defined as DEGs, and several Rubisco, SBPase, ALDase, TKase, and GAPDHase unigenes were highly expressed, with mean FPKM values greater than 100.0 (Appendix A). These results suggest these enzymes have significant regulatory functions affecting the CBC during *S. grandis* responses to grazing.

Rubisco catalyzes the carboxylation of the CO_2_ acceptor molecule ribulose 1,5-bisphosphate (RuBP) to initiate the CBC (Figure 2) [62]. This enzyme comprises eight large (rbcL) and eight small (rbcS) subunits [63], and its catalytic properties are determined by the large subunit encoded by the chloroplast genome [64]. In the current study, the five Rubisco DEGs were rbcS-encoding unigenes. Previous studies demonstrated that rbcS influences Rubisco catalytic efficiency, CO_2_ specificity, activity, quantity, assembly, and stability [65,66]. Moreover, rbcS and rbcL gene expression levels are positively correlated [67], and rbcS may function as a CO_2_ storage reservoir [68]. Thus, the highly expressed *S. grandis* rbcS unigenes (mean FPKM of 2156.42) identified in this study may indicate that regulating the Rubisco content is important for regulating the CBC in *S. grandis* as a response to differential grazing stresses.

RCA is an AAA^+^ ATPase that uses the energy from ATP hydrolysis to remove inhibitory sugars at the RCA site to generate a catalytically active enzyme with a temperature optimum below 40 °C [69,70]. In the present study, an RCA unigene (Unigene11360) was highly expressed under CK conditions (FPKM of 653.40), but with grazing, its expression was significantly down-regulated (Figure 2 and Appendix A), indicative of its varying roles under the four grazing conditions. Under CK conditions, the abundant RCA can accelerate CO_2_ fixation, activate Rubisco and induce the expression of key genes in the CBC [70]. However, the microenvironment of plants survive grazing changes because of a decrease in humidity and increases in temperature, surface exposure, light radiation, and evaporation, which ultimately lead to unstable and inactive RCA [71,72]. Additionally, because RCA is a labile protein in vivo, the cost of accumulating RCA is quite high [73]. Therefore, in response to grazing, *S. grandis* does not actively synthesize RCA, and the carbon turnover in the CBC is mediated by other CBC enzymes to maintain photosynthesis and regeneration.

Highly efficient photosynthetic CO_2_ fixation depends not only on the carboxylation capacity of Rubisco but also on the regeneration of RuBP [74]. This regeneration is largely regulated by SBPase, TKase, and ALDase [74], which catalyze the irreversible reactions and induce the metabolic branches of the CBC [40]. The over-expression of ALDase and SBPase genes individually or together in tobacco and *Arabidopsis thaliana* significantly increases photosynthetic activities as well as the overall biomass and seed yield, especially under elevated CO_2_ conditions [50,75]. However, a small decrease in the plastid TKase activity can dramatically inhibit photosynthesis and growth in antisense tobacco and cucumber transformants [76,77]. In the current study, ALDase, SBPase, and TKase unigenes in the *S. grandis* CBC were highly and differentially expressed (Figure 2 and Appendix A), suggesting that the transcriptional regulation and/or GEP of these enzymes may have important effects on RuBP regeneration, the photosynthetic capacity, and regrowth during *S. grandis* responses to grazing. Among these enzymes, ALDase and SBPase unigenes were similarly expressed (Figure 2 and Appendix A), indicating that, when photosynthesis was relatively strong under CK and HG conditions, the branching reaction efficiency of the CBC increased significantly in *S. grandis*. Consequently, ALDase effectively catalyzed the conversion of dihydroxyacetone phosphate (DHAP) and glyceraldehyde-3-phosphate (GAP) to fructose-1,6-bisphosphat (FBP) as well as the conversion of DHAP and erythrose 4-phosphate to sedoheptulose-1,7-bisphosphate (SBP) [78] (Figure 2), after which SBPase catalyzed the dephosphorylation of SBP to S7P (sedoheptulose-7-phosphate). These reactions can lead to the formation of a metabolic flux that enhances the carbon partitioning in the cycle and avoids the negative feedback regulation due to metabolic intermediates (e.g., glycolate and glyoxylate) [79,80]. Additionally, up-regulated ALDase and SBPase gene expression might further activate Rubisco by promoting the regeneration of RuBP in the CBC [50,81], thereby accelerating the carbon turnover to achieve compensatory photosynthesis and to stimulate the restorative growth of *S. grandis* plants. Interestingly, the expression of the TKase unigene (CL14956.Contig14) increased as the grazing intensity increased, with expression levels significantly higher than that under CK conditions. This suggests that the enzyme was actively engaged in regenerating RuPB in the grazed *S. grandis* plants and that it can effectively alleviate the limitation of RuBP to maintain photosynthesis under grazing stress. The significant up-regulation of TKase unigene expression might be highly related to the hyper-compensatory photosynthesis of *S. grandis*, especially under the LG and MG conditions.

### 3.3. Gene Expression Plasticity Affecting the Photorespiration Is Important for Stipa grandis Adaptations to Grazing

Plant photorespiration involves a complex network of enzymatic reactions and is linked to the CBC to form a photosynthetic photorespiratory super-cycle that is responsible for nearly all of the biological CO_2_ fixation on Earth [82]. Photorespiration begins with the oxygenation of RuBP by Rubisco, and the synthesized 2-phosphoglycolate (2-PG) causes a significant carbon loss and impedes carbon fixation as well as allocation [83]. Therefore, the degradation of 2-PG during photorespiration directly affects the overall carbon metabolism in plants [83].

The PGLPase enzyme catalyzes the formation of glycolate from 2-PG, and its activity is essential for plant carbon fixation and distribution. In this study, a PGLPase unigene (Unigene4982) was highly expressed in response to the CK and HG treatments, but its expression was significantly down-regulated under LG and MG conditions (Figure 3 and Appendix A). A previous study indicated that PGLPase expression and the 2-PG content are inversely related [83]. Therefore, the 2-PG levels under the LG and MG conditions were higher than those under the CK and HG conditions, reflecting the differences in carbon fixation and allocation between these treatments. Enzymatic analyses have demonstrated that high 2-PG levels inhibit *A. thaliana* triose-phosphate isomerase (TPI) and SBPase [83], and SBPase further limits the carbon flux of the RuBP regeneration phase in the CBC [84]. In *S. grandis*, PGLPase unigene expression was consistent with SBPase unigene expression, indicating RuBP regeneration was negatively regulated by 2-PG under LG and MG conditions. In contrast, under CK and HG conditions, the CBC carbon flux leading to RuBP regeneration was probably attributed to the substantial metabolism of 2-PG. Under heavy grazing conditions, abiotic stressors, such as high light intensity, water scarcity, increased temperatures, and elevated O_2_ partial pressures, might promote the oxidation of RuBP to form 2-PG [85]. The relatively high PGLPase unigene expression level (FPKM of 253.03) under this condition suggests that the PGLPase activity was not correlated with the photorespiratory flux (2-PG hydrolysis), and an increase in PGLPase activity may be beneficial for *S. grandis* under HG conditions. Hence, PGLPase is not a limiting factor for the photorespiratory flux, but it prepares the cycle for a considerable influx of 2-PG due to abiotic stresses [83,86] and enhances plant stress tolerance.

Another photorespiratory enzyme related to abiotic stress tolerance is GOXase, which catalyzes the conversion of glycolate to glyoxylate and produces H_2_O_2_. As a second messenger, H_2_O_2_ plays an important role in plant defense reactions [87]. Because it produces glyoxylate, GOXase represses Rubisco and RCA in rice and maize [88,89]. In *S. grandis*, the GOXase unigene (CL95.Contig34) expression pattern was similar to that of the RCA and Rubisco unigenes, reflecting the likely effect of GOXase on RCA and Rubisco. Nevertheless, under HG conditions, PGLPase and GOXase unigene expression levels were significantly up-regulated, which would have increased the tolerance of *S. grandis* to stressors due to grazing.

In the photorespiratory pathway, GGTase, SGTase, GDCase, and SHMTase form networks regulating glycine and serine (Figure 3). The plant cellular glycine: serine ratio is a sensitive indicator of photorespiratory activity [90]. The over-expression of GGTase genes can increase glycine and serine levels, whereas up-regulated SGTase gene expression has the opposite effect on serine levels [91]. These two reactions share common metabolites and exhibit a mutual decreasing trend (Figure 3). In this study, the unigenes encoding these two enzymes had the opposite expression patterns (GGTase: CL3941.Contig15; SGTase: CL593.Contig32), indicative of their roles in regulating the balance between glycine and serine contents in *S. grandis* under different grazing conditions. The GDCase gene expression level is reportedly a key determinant of photorespiratory flux control [92]. Remarkably, increases in GDCase activity facilitate carbon conversion throughout the photorespiratory cycle [93]. Simkin et al. [75] determined that plant growth and photosynthetic activities increase following the combined over-expression of GDC-H, SBPase, or ALDase genes. Therefore, the positive correlation between the photorespiration carbon flux and the CBC is one of the determinants of photosynthetic efficiency and biomass, indicating that adjusting the carbon flux via photorespiration to achieve compensatory photosynthesis is an important strategy adopted by *S. grandis* in response to differing grazing intensities. Here we construct a metabolic synthesis consistent with our data but not definitively proven by them, as a tentative understanding to allow the formulation of confirmatory hypotheses for testing in future research.

## 4. Materials and Methods

### 4.1. Study Site and Grazing Intensities

The grazing experiment was performed at the eastern edge of Xilinhot, Inner Mongolia, China (44°08′31″ N, 116°18′45″ E, Altitude 1129 m), characterized as a semi-arid continental climate with very cold and dry winters but warm and humid summers [94]. The average annual temperature in this region is 0 °C–4 °C, and the average precipitation is less than 300 mm. The rainfall is mostly concentrated in the June–September period (plant growth season stage). The soil type in this area is a ‘chestnut soil’ (i.e., calcic orthic Aridisol according to the US soil taxonomic system) [95], and the vegetation type is a typical steppe dominated by the perennial bunchgrass *Stipa grandis* and the perennial rhizomatous grass *L. chinensis*. Other companion species, such as *C. squarrosa*, *Agropyron cristatum*, and *Carex korshinskyi*, were also common in the study region. The grazing treatments were initiated in 2013. Specifically, 12 paddocks (120 m × 120 m) were fenced off, and four different grazing intensities were established in a randomized complete block design with three replicates. The four grazing intensities were as follows: no grazing (CK, no sheep), light grazing (LG, 2 sheep·ha^−1^), moderate grazing (MG, 4 sheep·ha^−1^), and heavy grazing (HG, 8 sheep·ha^−1^). For each grazing intensity, 28 Inner Mongolian Ujimqin sheep (3 years old, 60 kg body weight) were allowed to repeatedly graze for 3, 6, and 12 days per month for the LG, MG, and HG treatments (one after the other in series), respectively, during the vegetation growing season (i.e., June–September) every year. The sheep were allowed to graze from 7 am to 6 pm every day and were housed at night. They had free access to water and minerals.

### 4.2. Sample Collection, RNA Extraction, and Transcriptome Sequencing

In order to identify gene expression responses related to the compensatory photosynthesis behavior of *S. grandis* in the recovery growth stage after grazing, plant samples were collected from 9:00 am to 11:00 am on a sunny day at the end of July (two weeks after grazing). The steppe grasslands of this area have a net primary productivity in the region of 3–5 t dry matter (DM) ha^−1^ y^−1^, with July being the time of peak herbage accumulation rate in the mid growing-season [96,97]. Figure 5 shows the vegetation status of the grazing plots when the samples were collected. Emerging and healthy leaves of *S. grandis* plants were collected. In each of the four grazing intensities, three biological replicates were sampled. All samples were immediately frozen in liquid nitrogen and stored at −80 °C for the subsequent transcriptomic analysis. Total RNA was extracted from the frozen tissues with the TRIzol reagent (Invitrogen, Life Technologies Corporation, Carlsbad, CA, USA) according to the manufacturer’s instructions. The extracted RNA was treated with deoxyribonuclease I (TaKaRa Bio Inc., Otsu, Shiga, Japan) for 30 min at 37 °C to remove residual DNA. The total RNA was quantified, and the quality was assessed using the Agilent 2100 Bioanalyzer (Agilent Technologies, Palo Alto, CA, USA), with a minimum RNA integrity number of 6.5. The poly (A) mRNA was isolated with Oligo (dT) Beads and then used for the construction of cDNA libraries. Briefly, the purified mRNA was fragmented. Then first-strand cDNA was generated using random hexamer-primed reverse transcription, followed by second-strand cDNA synthesis. Afterward, A-Tailing Mix and RNA Index Adapters were added following end repair. The above cDNA fragments were amplified by PCR and purified by Ampure XP Beads. Then, the double-stranded PCR products were denatured and circularized with the splint oligo sequence to obtain a single-strand circle DNA (ssCir DNA). The ssCir DNAs were amplified with phi29 polymerase to make DNA nanoballs (DNBs) and to obtain the library. Finally, the cDNA library for each sample was sequenced paired-end with the BGISEQ-500 platform by BGI Tech Solution Co., Ltd. (Wuhan, China).

### 4.3. Data Filtering and de novo Assembly

The raw sequence reads for all samples were filtered with Trimmomatic (version 0.36) [98] to remove adapter-contaminated reads, low-quality reads (>20% low-quality nucleotides), and reads with ambiguous nucleotides (>5% ‘N’) to obtain clean reads, which were counted with SOAPnuke (version 1.4.0) [99]. The clean reads of each sample were then de novo assembled into a transcriptome using Trinity (version 2.0.6) [100], with an optimized k-mer length (25-mer). The subsequent clustering and elimination of redundancies were completed with TGICL (version 2.1) [101] to obtain unigenes. The unigenes assembled for all samples were then clustered with TGICL to obtain the non-redundant and unextendable assemblies (i.e., all-unigenes). To assess the completeness of the assembled transcriptomes, a BUSCO analysis was performed based on 303 conserved sequences in the eukaryotic database [102] (http://busco.ezlab.org/v2/datasets/eukaryota_odb9.tar.gz, accessed on 9 September 2018).

### 4.4. Gene Expression Quantification and Functional Annotation

Bowtie 2 (version 2.3.4.1) was used to map the reads of each sample to the merged transcriptome to quantify the expression level for each all-unigene in 12 samples [103]. The number of mapped reads was then estimated using RSEM (version 1.2.12) [104], with the default setting, for each sample. The normalized FPKM (fragments per kilobase of exon model per million mapped reads) values for each unigene in the 12 libraries were used to represent gene expression levels. To identify genes with reproducible expression levels in three biological replicates, we calculated the coefficient of variation (CV) for the gene expression of each grazing intensity. For each treatment, only genes with CV ≤ 0.5 were retained for further analyses. The correlations between all pairs of samples were analyzed with the hierarchical clustering of Pearson’s correlation coefficients based on the gene expression levels. Symmetrical heat maps were generated with ggplot2 (version 1.0.0) [105] within R version 3.0.2 (R Development Core Team, 2012). Next, a BUSCO analysis was performed to evaluate the completeness of the filtered transcripts. 

To predict the probable functions of the retained all-unigenes, their sequences were aligned with the sequences in public databases with the BLASTX algorithm, with a significance threshold E-value < 10^−5^. The following databases were screened: non-redundant (Nr) protein database (http://www.ncbi.nlm.nih.gov, accessed on 11 September 2018), the Swiss-Prot protein database (http://www.expasy.ch/sprot, accessed on 16 September 2018), the Kyoto Encyclopedia of Genes and Genomes (KEGG) pathway database (http://www.genome.jp/kegg, accessed on 16 September 2018), and the Cluster of Orthologous Groups (COG) database (http://www.ncbi.nlm.nih.gov/COG, accessed on 16 September 2018). The Blast2GO software (version 2.5.0) [106] and the Gene Ontology (GO) database were used to functionally annotate unigenes and assign them to the main GO functional categories (molecular function, cellular component, and biological process).

### 4.5. Identification of Differentially Expressed Genes (DEGs) and Analysis of Functional Enrichment

To identify DEGs among the four grazing intensities in the above dataset, pairwise comparisons were performed using an R package DEGseq (http://www.bioconductor.org/packages/release/bioc/html/DEGseq.html, accessed on 19 September 2018) [107]. The p-values were adjusted for multiple testing using the Benjamini-Hochberg method to control the false discovery rate (FDR) [108] and with Storey and Tibshirani’s statistical methods [109]. A FDR ≤ 0.001 and |log_2_ (fold-change)| ≥ 1 were used as the significance threshold to infer gene expression differences. To identify the most representative transcripts that might have played roles in response to grazing in *S. grandis*, we only considered DEGs that had mean FPKM values greater than 10.0 in all samples for further analysis. It is worth noting that this criterion might filter out a number of transcripts that were shut off or turned on in response to grazing stress because samples where these transcripts were not expressed might pull the mean FPKM under 10.0. To evaluate the expression patterns of the identified DEGs for the four grazing intensities, a K-means clustering analysis was performed by using the MeV software (version 4.9) [110]. The DEGs for each cluster were then subjected to GO and KEGG Ontology enrichment analyses by using MapMan (version 3.6.0) [111].

### 4.6. Experimental Validation of the Gene Expression Plasticity

*Stipa grandis* samples were collected for two consecutive years (2018 and 2019) from the plots mentioned in 4.2 for qRT-PCR analysis. Total RNA was isolated from the samples using RNAzol reagent (Genbetter, Beijing, China) according to the manufacturer’s instructions. The integrity and purity of the RNA samples were determined by 1% agarose gel electrophoresis and NanoDrop 2000 spectrophotometer (Thermo Fisher Scientific, Waltham, MA, USA). Subsequently, 1 ug total RNA was reverse transcribed into cDNA using a PrimeScript RT Reagent Kit with genomic DNA (gDNA) Eraser (TaKaRa, Dalian, China) according to the manual instructions. Nine unigenes in the Calvin–Benson cycle and photorespiration pathway were selected for qRT-PCR validation, including CL2153.Cotig7, Unigene48595, CL14956.Contig14, CL5196.Contig6, Unigene4982, CL95.Cotig34, CL3941.Contig15, CL1834.Conig3, and CL593.Contig32. Primers were designed using the NCBI Primer-Blast program (https://www.ncbi.nlm.nih.gov/tools/primer-blast/, accessed on 18 January 2021), and the primer sequences are shown in Appendix A. *Sgactin-7* was used as the internal reference gene [112]. qRT-PCR was carried out on a Eco real time PCR platform (Illumina, San Diego, CA, USA) using a SYBR-Green real-time PCR mix (TaKaRa, Dalian, China). Reactions were performed in a 20 ul final volume, which contained 100 ng of cDNA template, 0.8 ul (0.4 um) each of forward and reverse primers, 10 ul of SYBR Premix Ex Taq II, 0.4 ul of ROX Reference Dye or Dye II, and 6 ul of sterile distilled water. The thermal cycling programs were as follows: first denaturation 95℃ for 30 s, then 40 cycles of denaturation at 95 °C for 5s, annealing and extension at 60 °C for 34s. All reactions were performed on three technical replicates across the nine unigenes. The relative expression levels of the investigated genes were normalized to *Sgactin-7* and calculated using the 2^−ΔΔCt^ method.

## 5. Conclusions

Comparative transcriptomic analyses revealed that the gene expression of *S. grandis* has plasticity induced by long-term differential grazing intensities, which involves altered regulation of many biological processes and metabolic pathways. The similar expression profiles of genes related to the CBC and photorespiration pathways suggest that these pathways synergistically respond to grazing to promote recovery growth and stress tolerance in *S. grandis*. Our findings provide novel insights into the grazing responses of *S. grandis* on the gene expressional level and will facilitate future investigations of the relevant regulatory roles and mechanisms of genes underlying the plastic response of grassland plant species to grazing.

## Figures and Tables

**Figure 1 ijms-22-11882-f001:**
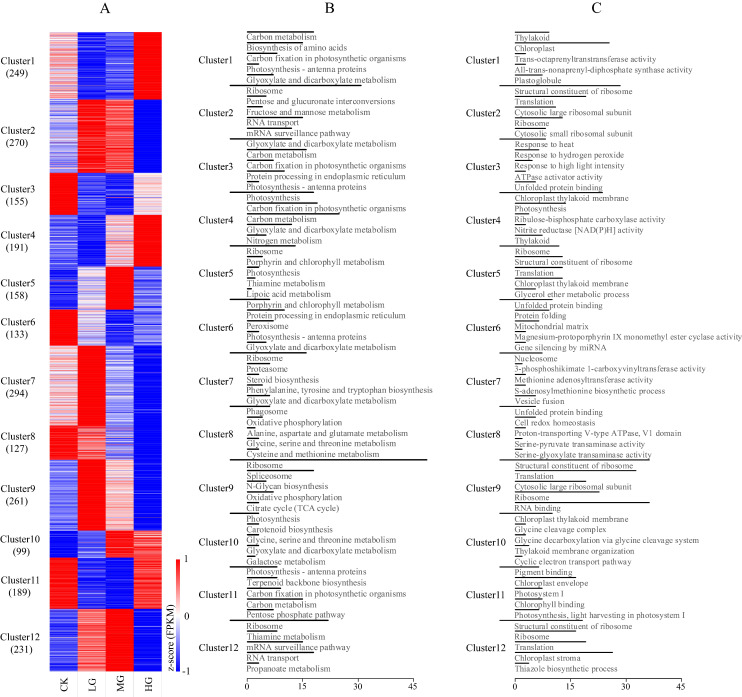
Expression patterns and functional annotations of the differentially expressed genes (DEGs). (**A**) K-means clustering of DEGs. The red-to-blue gradient indicates high-to-low expression levels. Prior to clustering, the expression level for each transcript over the four grazing treatments was standardized using z-scores. (**B**) KEGG pathway enrichment analysis. (**C**) GO enrichment analysis. In panels B and C, only the top five enriched GO terms and KEGG pathways are shown, respectively; the x-axis indicates the number of the unigenes, and clusters are separated by left-extended short black bars.

**Figure 2 ijms-22-11882-f002:**
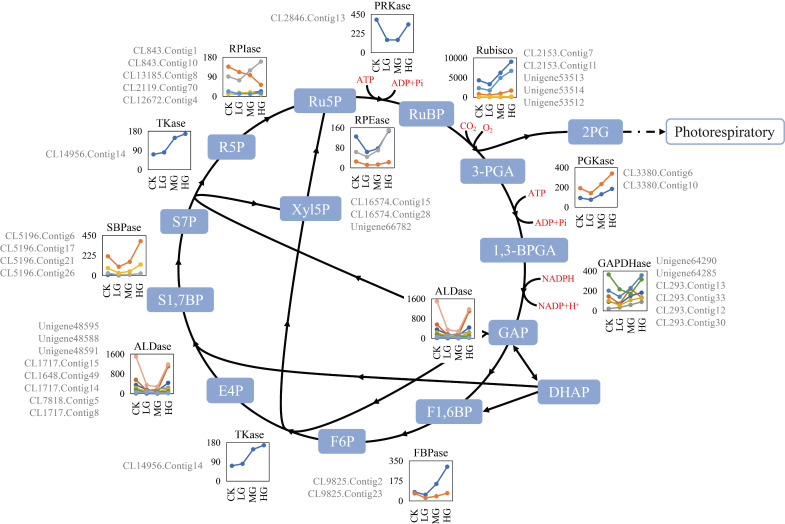
Expression patterns of differentially expressed genes (DEGs) in the Calvin–Benson cycle (CBC). The line and symbol chart next to each enzyme represents the expression profiles of DEGs shown in Appendix A. The lines with different colors represent the assembled transcripts for an enzyme in the CBC. According to the gene expression levels under CK conditions, these transcripts were listed next to each chart in descending order. The grazing gradient is shown on the x-axis, and the gene expression level (the mean FPKM value of three biological replicates) is shown on the y-axis. The carboxylation reaction catalyzed by Rubisco fixes CO_2_ into the acceptor molecule RuBP, forming 3-PGA. The reductive phase of the cycle follows with two reactions catalyzed by PGKase and GAPDHase, producing GAP. The GAP enters the regenerative phase catalyzed by ALDase and either FBPase or SBPase, producing F6P (fructose-6-phosphate) and S7P (sedoheptulose-7-phosphate). The F6P and S7P are then used in reactions catalyzed by TKase, RPIase, and RPEase, producing Ru5P (ribulose 5-phosphate). The final step, which is catalyzed by PRKase, converts Ru5P to RuBP. Rubisco is the initiating enzyme for the Calvin–Benson cycle and the photorespiratory cycle, fixing O_2_ into the acceptor molecule RuBP to form 2-PG, which is then metabolized via the photorespiratory pathway.

**Figure 3 ijms-22-11882-f003:**
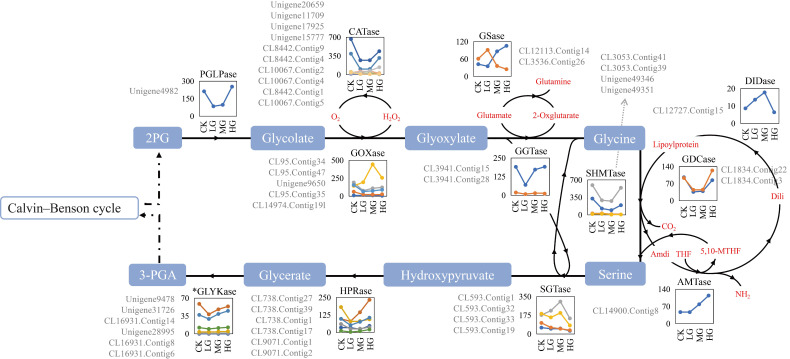
Expression patterns of differentially expressed genes (DEGs) in the photorespiratory pathway. The line and symbol chart next to each enzyme represents the expression profiles of DEGs shown in Appendix A. The lines with different colors represent the assembled transcripts for an enzyme in the photorespiratory pathway. According to the gene expression levels under CK conditions, these transcripts were listed next to each chart in descending order. The grazing gradient is shown on the x-axis and the gene expression level (the mean FPKM value of three biological replicates) is shown on the y-axis. The photorespiratory cycle is a process in photosynthetic cells involving the chloroplasts, peroxisomes, mitochondria, and the cytosol. In chloroplasts, Rubisco catalyzes the oxygenation of RuBP, which generates one molecule of 3-PGA and one molecule of 2-PG. The 2-PG is first dephosphorylated to glycolate by PGLPase, after which it diffuses into the peroxisome. In the peroxisome, the O_2_-dependent glycolate is oxidized to glyoxylate by GOXase to produce H_2_O_2_, which is quickly detoxified by CATase. Glyoxylate is transaminated to glycine by the parallel action of GGTase or SGTase. Glycine then moves into the mitochondrion, wherein the GDCase multienzyme system and SHMTase convert two molecules of glycine to one molecule of serine, NH_3_, and CO_2_. After being transported from the mitochondrion to the peroxisome, serine is converted by SGTase to hydroxypyruvate, which is reduced to glycerate by HPRase. The glycerate returns to the chloroplast to be phosphorylated by GLYKase (glycerate 3-kinase), and the resulting 3-PGA is converted to RuBP in the Calvin–Benson cycle. * represents transcripts that were not DEGs among the four grazing treatments.

**Figure 4 ijms-22-11882-f004:**
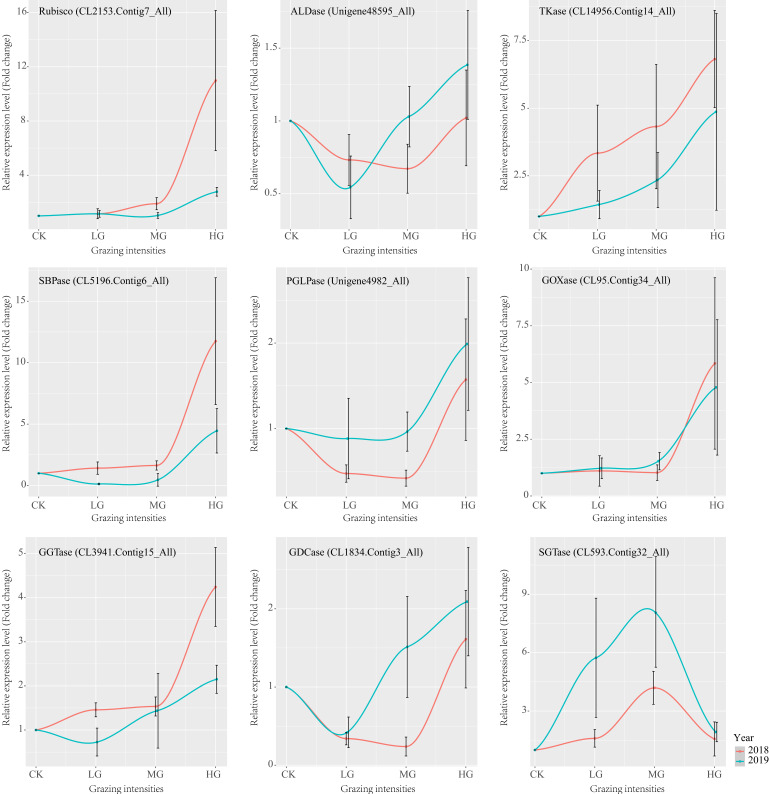
Expression patterns of selected genes measured using qRT-PCR. Nine genes were selected for validating observations of gene expression plasticity in various grazing intensities in two years of *S. grandis* samples. The x-axis represents four different grazing intensities, and the y-axis indicates fold change of genes’ relative expression levels. The color curves represent the gene expression patterns of the selected genes in 2018 and 2019, respectively. The error bars represent mean standard deviations (± SD) of three biological replicates.

**Figure 5 ijms-22-11882-f005:**
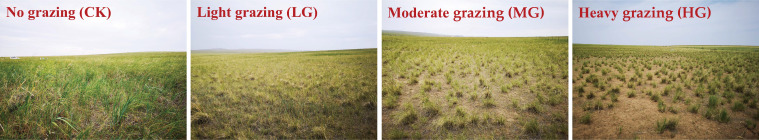
Vegetation status of plots of the four grazing intensity treatments (Photographed 28 July 2018).

**Table 1 ijms-22-11882-t001:** Summary of sequencing and assembly results.

Sample	RR (M)	CR (M)	GCC (%)	Q20 (%)	AU (No.)	ML (bp)	N50 (bp)	TMR (%)
CK_1	106.66	101.80	47.79	97.73	101,788	1373	2078	91.72
CK_2	105.80	101.64	47.57	97.59	101,392	1322	1953	92.22
CK_3	104.76	100.89	47.81	97.83	89,023	1363	2030	93.14
LG_1	105.96	102.07	47.62	97.72	108,636	1300	1890	92.27
LG_2	107.36	102.71	47.92	98.03	113,212	1327	1956	90.52
LG_3	105.71	101.88	47.95	97.78	115,918	1373	2052	91.75
MG_1	107.08	102.77	47.42	98.12	67,705	906	1243	89.19
MG_2	107.54	101.84	47.51	97.75	103,648	1304	1917	92.73
MG_3	105.94	101.38	47.51	97.94	107,998	1304	1919	91.35
HG_1	105.73	101.37	47.37	97.89	70,978	1144	1636	93.49
HG_2	106.00	102.09	47.00	98.20	88,736	1239	1806	92.09
HG_3	106.01	102.90	47.07	98.43	90,147	1138	1644	92.28
All-Unigene	—	—	47.27	—	251,412	1854	2536	—

RR and CR denote raw read and clean read, respectively. M represents million. GCC represents GC content. Q20 means the percentage of bases with a Phred value >20. AU represents the number of assembled unigenes. ML indicates the mean length of the assembled sequences. N50 represents 50% of the assembled bases that were incorporated into sequences with length of N50 or longer. TMR indicates total mapped clean reads to an assembled transcriptome.

## Data Availability

The datasets supporting the conclusions of this article are available in the NCBI Sequence Read Archive (SRA) repository under the project name PRJNA658710 (https://www.ncbi.nlm.nih.gov/sra/?term=PRJNA658710, accessed on 3 October 2021). All other datasets generated for this study are included in the article/Appendix A, further inquiries can be directed to the corresponding authors.

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
