# Peer review of "Transcriptome-Wide Gene Expression Plasticity in Stipa grandis in Response to Grazing Intensity Differences"

_ijms, 2021, doi:10.3390/ijms222111882_

Round 1
Reviewer 1 Report
This manuscript brings a lot of original RNAseq data and its conclusions concern interesting topic - adaptation of plants to herbivory. The methods used for this research are in principle adequate and the authors used also qPCR to confirm the results deduced from RNAseqs.
There are however also some points for improvement:
1.) FPKM normalization is currently taken as problematic for this kind of research as few highly expressed genes can cause significant bias if several FPKM normalised samples are compared. More suitable approaches for normalization of RNAseq data have been developed and they can be easily implemented using, e. g., edgeR package in R.
2.) The authors should refer to and discuss paper: Wan D, Wan Y, Hou X, Ren W, Ding Y, Sa R. De novo assembly and transcriptomic profiling of the grazing response in Stipa grandis. PLoS One. 2015 Apr 13;10(4):e0122641. doi: 10.1371/journal.pone.0122641. PMID: 25875617; PMCID: PMC4395228.
The approach used in this paper is very similar but with just two types of samples compared. The details of analyses and also conclusions are different.
3.) The form of the last reference is not correct as there is a switch in the names and the surnames of the authors
Instead of:
Dongli, W.; Yongqing, W.; Qi, Y.; Bo, Z.; Weibo, R.; Yong, D.; Zhen, W.; Ruigang, W.; Kai, W.; Xiangyang, H. Selection of 941
Reference Genes for qRT-PCR Analysis of Gene Expression in Stipa grandis during Environmental Stresses. PLoS One 2017, 942
12, e0169465.
there should be:
Wan D, Wan Y, Yang Q, Zou B, Ren W, Ding Y, et al. (2017) Selection of Reference Genes for qRT-PCR Analysis of Gene Expression in Stipa grandis during Environmental Stresses. PLoS ONE 12(1): e0169465. https://doi.org/10.1371/journal.pone.0169465
Reviewer 2 Report
The authors present their study on how grazing of the steppe grass Stipa grandis affects gene expression. They took samples from paddocks that had been exposed to no, low, medium, and high grazing by sheep. From these samples they extracted RNA and performed RNA-Sequencing in order to assemble a transcriptome that then could be used for gene expression analyses. In their differential expression results, the Calvin-Benson cycle together with the photorespiratory pathway emerged as being significantly affected by grazing. The authors proceed to conclude that the effects on these two pathways might be synergistic, and results in improves stress tolerance.
While I find the setup of the experiment interesting and useful, I do have some concerns regarding this study. To start with, this is not the first time that the transcriptional response to grazing in plants has been characterised. Wan et al. (2015; https://doi.org/10.1371/journal.pone.0122641) looked at transcriptomic profiling in Stipa grandis, and while I consider their experimental setup inferior to that of the study at hand (only two conditions, no replication), I still think it would be worth mentioning. There are also studies in other species that might be interesting to discuss. For example, Wang et al. (2016; https://doi.org/10.1038/srep19438) looked at the transcriptomic grazing response in alfalfa (Medicago sativa) and found that pathways related to e.g. translation, carbon assimilation, and pathogen defence response were maintained at a more functional level in their grazing tolerant population. It would have been interesting to see the authors discuss similarities and differences with this study. Since I am not very familiar with the field, there might be more interesting research that could be discussed in the context of this study.
My main concern with this work relates to the gene expression analyses. The methods are not detailed enough to properly assess what has been done. There are also parts that are ambiguous, or contradict a previous statement. These issues must be addressed before this manuscript can be considered for publication. Therefore, I will not go into detail regarding the results and the conclusions, but rather focus mostly on the technical aspects.
- L134-137: Was the BUSCO analysis only performed on the transcriptomes prior to filtering? If that is the case, it would be very informative to see how it looked after the filtering as well, i.e. on the 33,241 unigenes.
- Table 1: Two of the samples (STIMG_2 and STIMG_3) have the same number of reads after filtering. Is this correct? It just looks a bit strange considering all other samples lost a few million reads. Also, is TMR a percentage of the raw reads or the clean reads?
- L145: I have not heard the term coefficient of variance before. Should this be coefficient of variation, i.e. standard deviation divided by the mean?
- L155: Figure S2 is redundant in my opinion. The text contains all the information that is displayed in the figure.
- L165-166: If I interpret this sentence correctly, only a subset of genes was used for the differential expression analysis. While it sometimes can be useful to exclude genes with very low expression for memory reasons, I don't think that 28,540 had such low expression to warrant their removal. Furthermore, tools such as DESeq2 use information from all genes to estimate the dispersion, so excluding a large number of genes prior to analysis might affect the results.
- L170-171: Here it says that the genes were clustered using K-means clustering, while in the legend of Figure 1 it says hierarchical clustering. Which is it?
- Figure 1: For panel a, it is stated that "red-to-blue gradient indicates high-to-low expression levels", but the scale goes from 0 to 1. Have the expression values been scaled? Also, if the scale itself is not centered around zero (e.g. z-score), I find that a sequential colour palette is to prefer to a diverging palette.
- Table 2: The table header has been split onto two lines. If the header does not fit on a single line, then consider putting a line break in a more appropriate position, alternatively have a nested table header, something like this:
As a final note on this table, I am not sure that this is the best way of presenting these results. Figure 2 seems to contain just about the same information (apart from the fold changes), and I would be perfectly happy to only have this figure in the main text and perhaps have this table as a supplement, maybe integrated with Table S6.FPKM log2 fold change Gene ID CK LG MG HG LG/CK MG/CK HG/CK - Table 3: The same comments as for Table 2 apply.
- L649-651: This is related to one of my previous comments. I find this criterion a little bit strange. What about genes that are shut off completely? Or genes that are turned on as a response to grazing? Samples where these genes are not expressed might pull the mean FPKM under 10, but I would argue that these might be some of the more interesting examples. Also, from this section I get the impression that the filtering was done after the differential expression analysis, while in the results it sounds like it was performed prior it. The discussion (L349-353) also suggest that the filtering was done prior to the differential expression analysis. Which one is it?
- L623: I have not used RSEM for quite some time, but looking at Github (https://github.com/deweylab/RSEM/releases), it seems that version 1.2.8 was released in 2013. Was there a particular reason for using such an old version? Also, I noticed that Bowtie 2 support wasn't added until version 1.2.11, but I don't know if this has any practical implication.
- Speaking of the gene expression quantification, FPKM is really a inferior metric to use. It is deeply flawed, and should be replaced by something like transcripts per million (TPM; https://doi.org/10.1186/1471-2105-12-323, https://doi-org.proxy.ub.umu.se/10.1007/s12064-012-0162-3) or trimmed mean of M-values (TMM; https://doi.org/10.1186/gb-2010-11-3-r25). Zhao et al. discuss this issue in their review: https://doi.org/10.1261/rna.074922.120. They also make the case that TPM can be misused if there is RNA composition bias, but in this particular study it would be a much better choice than FPKM from what I can tell. As they state, FPKM is not reliably comparable between samples, even in the same study, and therefore I am forced to take all results from the gene expression analyses with a grain of salt. The issue is also mentioned by the trinity manual (https://github.com/trinityrnaseq/trinityrnaseq/wiki/Trinity-Transcript-Quantification#estimating-transcript-abundance).
- L645: Did the authors use DESeq? The reference suggests that DEGSeq was used, but I cannot access the link in this sentence (HTTP 403 Forbidden). If the R package DEGseq was used, I would say that this is not the best choice for this analysis. From the paper, it looks like it uses a Poisson distribution for modelling read counts, and it has since been established that this is prone to produce a lot of false positives, and that something like a negative binomial distribution is more appropriate. Also, it is not clear how the differential expression analysis was performed. What was the input to DEGseq (or DESeq)? What was the model that was used for the different comparisons? It is critical that these details are provided in the manuscript.
- It is not clear to me what sequencing protocol was used. Single-end or paired-end? Strand-specific or not? These details need to be presented.
Best regards
Niklas Mähler
Round 2
Reviewer 2 Report
The authors have addressed most of the comments I had for the first version of the manuscript. I particularly approve of the changes to the methodology which makes it much easier to follow what has been done, and I also like that a caveat regarding the differential expression analysis was added to the discussion. I am however still not approving of the use of FPKM and DEGseq for the transcriptomic analyses. I very much appreciate that the authors do not want to redo large parts of their analysis in order to address this potential issue, but at the same time their arguments frankly fall flat.
It has been shown on numerous occasions that DEGseq is inferior to other software packages for this type of analysis. Here are just a few examples:
- Schurch et al. 2016 (https://rnajournal.cshlp.org/content/22/6/839.full.html): "DEGSeq and NOISeq both show strong TPR performance but this is coupled with high FPRs (DEGSeq: ∼17%, NOISeq: ∼9%). For DEGSeq in particular this originates from overestimating the number of SDE genes regardless of the number of replicates." and "DEGSeq, in particular, has poor false positive performance with every bootstrap iteration identifying >5% of all genes as false positives (FPs) and a median FPR of ∼50% irrespective of the number of replicates."
- Guo et al. 2013 (https://link.springer.com/article/10.1186/1471-2164-14-S8-S2): "Out of 16,146 genes, DEGseq identified the most number of significant genes at 15,226. That is, 94.3% of all genes were identified by DEGseq as statistically significantly differentially expressed between tumor and normal, which implies that DEGseq is over-sensitive." and "Among all six methods, baySeq has the smallest FPR, and DEGseq has the largest FPR."
- Froussios et al. 2019 (https://academic.oup.com/bioinformatics/article/35/18/3372/5307752): "DEGseq fails to control its FP fraction adequately, likely due to over-estimation of the number of significantly differentially expressed genes."
If we want to compare citation metrics, then DESeq2 has 314,000 article accesses and 19,948 citations in Web of Science since 2014 (https://genomebiology.biomedcentral.com/articles/10.1186/s13059-014-0550-8/metrics), and edgeR has over 83,000 views and over 19,000 citations since 2010 (https://oxfordjournals.altmetric.com/details/1130896). My point is that there might not be a consensus on what is the single best method, but there is definitely a consensus on FPKM and DEGseq not being the best methods. At least the authors acknowledge this in their updated text, but the study would have been so much stronger and more interesting with a more robust analysis of this data.
A couple of minor comments are listed below.
- The colour scale in Figure 1a should be sequential, i.e. a palette with decreasing luminance as values get higher. Also, what does it mean that the expression values were scaled to the interval [0, 1]? If there is a single very high value, then the remaining values would be pushed close to zero. This does not seem to be the case just from looking at the figure, but I would say that a z-score standardisation together with a diverging colour scale (like the one currently in use) would be preferred for this visualisation.
- In Figure 4, given that there are only three replicates for each point, there is nothing preventing the authors from showing all the data points in the figure. This would give the reader a better understanding of the variation in this data. Furthermore, I would have preferred to see the different panels plotted on the same scale. Right now it looks like all of them have very dramatic changes until you start looking at the scales of the individual panels.
